

# Discrete natural neighbour interpolation with uncertainty using cross-validation error-distance fields

Thomas R. Etherington

Manaaki Whenua—Landcare Research, Lincoln, New Zealand

## ABSTRACT

Interpolation techniques provide a method to convert point data of a geographic phenomenon into a continuous field estimate of that phenomenon, and have become a fundamental geocomputational technique of spatial and geographical analysts. Natural neighbour interpolation is one method of interpolation that has several useful properties: it is an exact interpolator, it creates a smooth surface free of any discontinuities, it is a local method, is spatially adaptive, requires no statistical assumptions, can be applied to small datasets, and is parameter free. However, as with any interpolation method, there will be uncertainty in how well the interpolated field values reflect actual phenomenon values. Using a method based on natural neighbour distance based rates of error calculated for data points via cross-validation, a cross-validation error-distance field can be produced to associate uncertainty with the interpolation. Virtual geography experiments demonstrate that given an appropriate number of data points and spatial-autocorrelation of the phenomenon being interpolated, the natural neighbour interpolation and cross-validation error-distance fields provide reliable estimates of value and error within the convex hull of the data points. While this method does not replace the need for analysts to use sound judgement in their interpolations, for those researchers for whom natural neighbour interpolation is the best interpolation option the method presented provides a way to assess the uncertainty associated with natural neighbour interpolations.

## INTRODUCTION

Spatially continuous geographic phenomena are often only measured at point locations. Interpolation techniques provide a method to convert such point data into a continuous estimate of the phenomenon, and have become a fundamental computational technique of spatial and geographical analysts with key texts devoting large sections to interpolation methods (*Burrough & McDonnell, 1998*; *O'Sullivan & Unwin, 2010*; *Slocum et al., 2014*).

Natural neighbour (or Sibson) interpolation is an interpolation technique that was first presented by *Sibson (1981)*. The method is based upon a Voronoi (or: Dirichlet, Thiessen) diagram that partitions space to identify those areas that are closest to a set of points (*Okabe et al., 2000*). Previous authors (*Sambridge, Braun & McQueen, 1995*; *Watson, 1999*) have noted several useful properties of natural neighbour interpolation: (i) the method is an

Corresponding author
Thomas R. Etherington, etheringtont@landcareresearch.co.nz

exact interpolator, in that the original data values are retained at the reference data points; (ii) the method creates a smooth surface free from any discontinuities; (iii) the method is entirely local, as it is based on a minimal subset of data locations that excludes locations that, while close, are more distant than another location in a similar direction; and (iv) the method is spatially adaptive, automatically adapting to local variation in data density or spatial arrangement. To this list I would add: (v) there is no requirement to make statistical assumptions; (vi) the method can be applied to very small datasets as it is not statistically based; and (vii) the method is parameter free, so no input parameters that will affect the success of the interpolation need to be specified.

These properties make natural neighbour interpolation particularly well suited for the interpolation of continuous geographic phenomena from data points that have a highly irregular spatial distribution. While the choice of an appropriate interpolation method will always vary on a case by case basis, studies comparing interpolation methodologies with climate and land surface data demonstrate that natural neighbour interpolation is a highly competitive and sometimes optimal technique (*Abramov & McEwan, 2004*; *Bater & Coops, 2009*; *Hofstra et al., 2008*; *Lyra et al., 2018*; *Yilmaz, 2007*).

Unfortunately, natural neighbour interpolation can be relatively slow in comparison to other methods (*Abramov & McEwan, 2004*). The high computational cost arises from the need to insert a new point into the Voronoi diagram for every cell that will make up the interpolation field, and this geometric process becomes increasingly difficult in higher dimensions (*Park et al., 2006*). This has led to the development of discrete (or digital) natural neighbour interpolation that is significantly quicker than traditional approaches (*Park et al., 2006*) and has been applied successfully in a geographical context (*Keller et al., 2015*).

While natural neighbour interpolation has various useful properties, and the discrete form is computationally scalable, there is a great deal of uncertainty associated with any interpolation. Therefore, being able to associate interpolation estimates with some form of uncertainty would be highly desirable. Previous efforts for natural neighbour interpolation have been based upon fitting statistical uncertainty models (*Bater & Coops, 2009*; *Ghosh, Gelfrand & Mlhave, 2012*), but this approach is contrary to natural neighbour interpolation's useful properties (v), (vi), and (vii). Therefore, for those researchers who decide that for their data and objectives natural neighbour interpolation is the best interpolation option, I present an approach to associate the interpolation with a measure of uncertainty that is consistent with all the useful properties of natural neighbour interpolation.

## MATERIALS & METHODS

### Discrete natural neighbour interpolation

In the 2-dimensional planar context that is most relevant to geographical applications, discrete natural neighbour interpolation begins by calculating a discrete Voronoi diagram. First, a raster spatial domain $C$ of cells $c$ is defined such that $c \in C \subset \mathbb{R}^2$ and hence each $c$ has coordinate attributes $x, y$ for its centre so all $c_i = \{x_i, y_i\}$.

The data points are then used to define a set $P$ of $n$ data cells $P = \{p_1, p_2, p_3, \ldots, p_n\}$ where $P \in C$, and each data cell has coordinate attributes for its cell centre $x, y$ and value $z$, so $p_i = \{x_i, y_i, z_i\}$. When multiple data points occur within a raster cell, the resulting data cell has a value $z$ that is the mean of all the data point values.

The discrete Voronoi polygon $V(p_i)$ that contains all the cells that are closest to each data cell can then be defined as

$$V(p_i) = \{c \in C | d(c \to p_i) < d(c \to p_j) \; \forall \, j \neq i\} \tag{1}$$

where $d(c \to p)$ is the Euclidean distance between the centre of the cells $c$ and $p$. When $c$ is equally distant from more than one $p$ for convenience $c$ is assigned to the $p$ with smallest index. The set of $n$ discrete Voronoi polygons then creates the discrete Voronoi diagram

$$V(P) = \{V(p_1), V(p_2), V(p_3), \ldots, V(p_n)\} \tag{2}$$

that identifies which raster cells are closest to which data cells (Fig. 1A) (*Okabe et al., 2000*). In the process of calculating $V(P)$ another set $D(P \to C)$ that records the Euclidean distance from the set of data cells $P$ to all raster cells $C$ (Fig. 1B) is created. As each data cell $p_i$ has an associated value $z_i$, $V(P)$ can be used to interpolate the data cell values across the raster to produce $Z(P)$, which in a geographic information system (GIS) context is equivalent to nearest neighbour interpolation (*Burrough & McDonnell, 1998*; *Tomlin, 1990*) (Fig. 1C).

To interpolate the data cell values using natural neighbour interpolation, the set of Euclidean distances from an interpolation cell $c_i$ to all raster cells $D(c_i \to C)$ is calculated (Fig. 1D). Then the discrete Voronoi polygon for the interpolation cell $V(c_i)$ is defined as

$$V(c_i) = \{c \in C | D(c_i \to C) \leq D(P \to C)\} \tag{3}$$

that is the set of raster cells that are as close or closer to the interpolation cell than any data cell. The set $V(c_i)$ can then be used to find the set of relevant data cell values

$$Z(c_i) = \{c \in Z(P) | c \in V(c_i)\} \tag{4}$$

that will form the basis on the interpolation to that cell (Fig. 1E). The natural neighbour interpolation estimate $\hat{z}$ is then calculated as

$$\hat{z}(c_i) = \frac{\sum Z(c_i)}{\sharp Z(c_i)} \tag{5}$$

where $\sum_Z(c_i)$ is the sum of the cell values in $Z(c_i)$ and $\sharp Z(c_i)$ is the number of cells in the set $Z(c_i)$, hence $\hat{z}(c_i)$ is simply the mean of $Z(c_i)$. By calculating $\hat{z}(c_i)$ for all raster cells the natural neighbour interpolation is produced (Fig. 1F).

## Calculating uncertainty
### Cross-validation error
Global error estimation is a traditional approach to measure the uncertainty of geographic models (*Zhang & Goodchild, 2002*). Given a set of $n$ paired observed $o$ and modelled $m$ values, the absolute error $e_i$ for each pair is $e_i = |m_i - o_i|$, and a global estimate of error using a method such as the mean absolute error (MAE) is calculated as

$$MAE = \frac{1}{n} \sum_{i=1}^{n} e_i \tag{6}$$

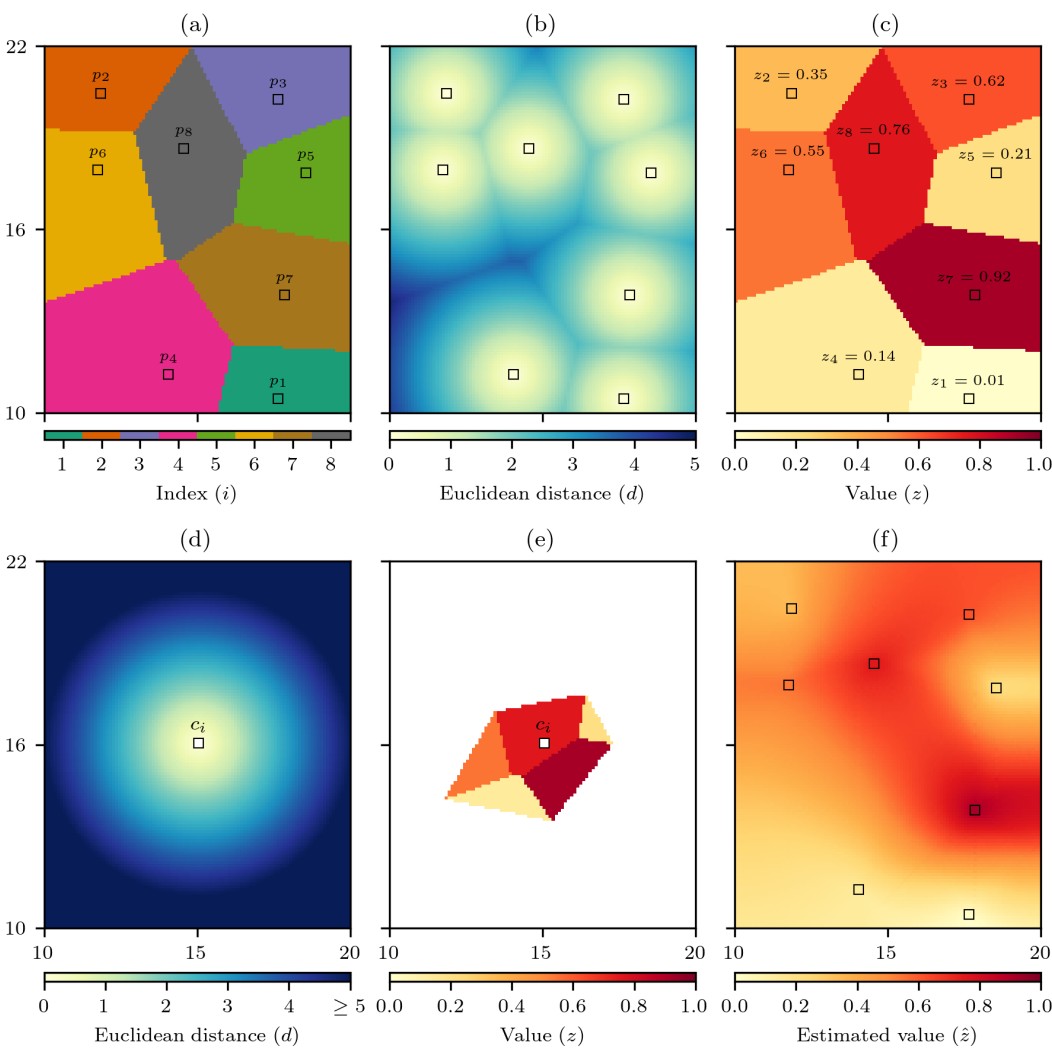

**Figure 1   Discrete natural neighbour interpolation.** (A) For a set $P$ of $n$ data cells $p$ the discrete Voronoi diagram $V(P)$ defines which raster cells are closest to which data cells and (B) the distance to the closest data cell $D(P \rightarrow C)$. (C) $V(P)$ is used to interpolate the values $z$ of the data cells to produce $Z(P)$. (D) For an interpolation cell $c_i$ the distance to all raster cells $C$ is calculated as $D(c_i \rightarrow C)$, and (E) by comparing $D(c_i \rightarrow C)$ to $D(P \rightarrow C)$ identifies $Z(c_i)$ which are those cells of $Z(P)$ that are as close or closer to the $c_i$ than any data cell $p$. The mean value of $Z(c_i)$ is the natural neighbour interpolation estimate $\hat{z}$ for $c_i$, and by repeating this process for all raster cells (F) the natural neighbour interpolation is produced.

that is simply the mean of all the absolute errors (*Willmott & Matsuura, 2005*).

However, there is little point in doing this for the data cells of natural neighbour interpolation as given property (i) that it is an exact interpolator the estimated value $\hat{z}_i$ for the data cells will always be the same as the actual value $z_i$ so the absolute errors will always be zero. Therefore, MAE needs to be applied in conjunction with a cross-validation approach that iteratively withholds each data cell $p_i$ from the set of data cells $P$ to produce the set $\{P - p_i\}$, and then uses interpolation to estimate the value $\hat{z}_i$ at the withheld data cell $p_i$ on the basis of a discrete Voronoi diagram $V(\{P - p_i\})$ that is developed without the

withheld data cell. The absolute error $e_i$ for each data cell $p_i$ is then calculated as $e_i = |\hat{z}_i - z_i|$ and the *cross-validation MAE* can be calculated using Eq. (6).

Even with cross-validation the MAE like all global error estimates, such as the commonly used root-mean-square error (RMSE), are not ideal measures of uncertainty for a spatial interpolation (*Zhang & Goodchild, 2002*). As non-spatial methods that average errors across space they cannot indicate if errors are consistent across space or if higher errors in one region are balanced out by lower errors in another region. This is a critical limitation of global error estimation methods, as for application purposes it could be very useful to know where the interpolation uncertainty is higher or lower.

### Cross-validation error field

One way to communicate the spatial uncertainty of geographical information is to map estimates of error (*Zhang & Goodchild, 2002*). This has been attempted before for natural neighbour interpolation (*Bater & Coops, 2009*; *Ghosh, Gelfrand & Mlhave, 2012*), but as already noted these statistical modelling approaches are contrary to natural neighbour interpolation's useful properties (v), (vi), and (vii).

Another way to map estimates of error that is consistent with the properties of natural neighbour interpolation is the *cross-validation error field* (*Willmott & Matsuura, 2006*). This process begins in a similar manner to the cross-validation MAE, but once $e$ has been calculated for each data cell, rather than average the errors using Eq. (6) the errors are assumed to be spatially autocorrelated and interpolation is used to interpolate $e$ to estimate an absolute error field $\hat{e}$. This use of localised absolute errors is highly advantageous as it is consistent with property (iii) of natural neighbour interpolation and allows for error estimates to reflect local changes in the spatial-autocorrelation of the phenomenon being interpolated, with lower errors in more autocorrelated areas and higher errors less autocorrelated areas.

However, while the cross-validation error field does indicate where interpolation errors are likely to be higher, it cannot be used directly as a measure of uncertainty for natural neighbour interpolation as ultimately the interpolation is calculated using all $n$ data cells and given property (i) of natural neighbour interpolation is that it is an exact interpolator we know we will have zero error and hence zero uncertainty at the data cells.

On the basis of Tobler's first law of geography that "everything is related to everything else, but near things are more related than distant things" (*Tobler, 1970*), *Zhang & Goodchild (2002)* recognise that distance is an important component of uncertainty as locations nearer to data should have less uncertainty. This relationship of increasing error with increasing distance to data has even been demonstrated for natural neighbour interpolation (*Keller et al., 2015*). Therefore, I propose to extend the cross-validation error field idea by incorporating distance to produce a *cross-validation error-distance field* that will better represent the uncertainty associated with natural neighbour interpolation.

### Natural neighbour distances

A positive relationship between natural neighbour interpolation absolute errors and the minimum distance to a data cell has been shown (*Keller et al., 2015*), so this relationship could be used to predict absolute error as a function of distance from the nearest data

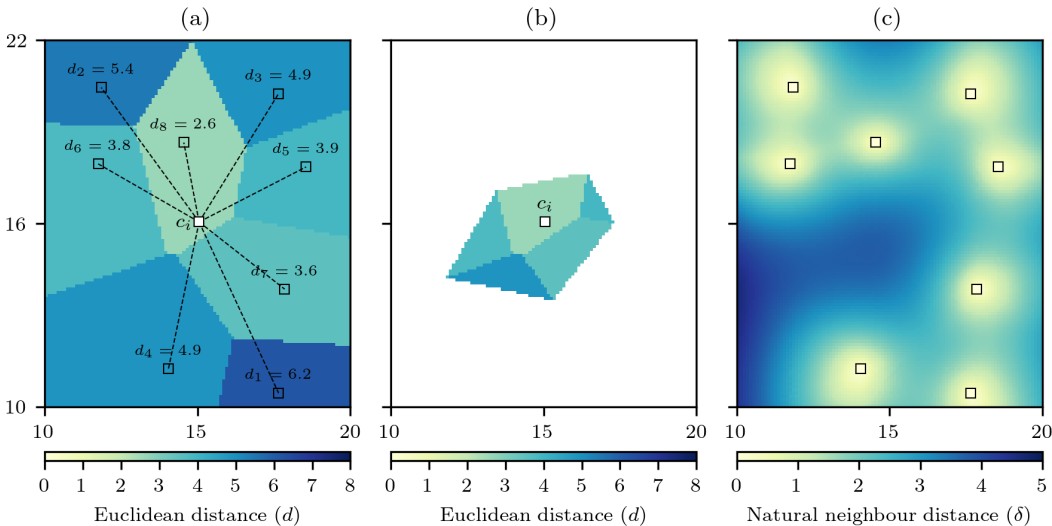

**Figure 2  Computation of the natural neighbour distance.** (A) For an interpolation raster cell $c_i$ the Euclidean distance to all data cells $d_j$ is calculated, and the discrete Voronoi diagram $V(P)$ is used to produce $D(P)$ that interpolates the distances by the discrete Voronoi polygons. (B) the cells of $D(P)$ that are closer to $c_i$ than any data cell defines the set $D(c_i)$ and the mean value of this set gives the natural neighbour distance $\delta$ for $c_i$. (C) When repeated for all raster cells a natural neighbour distance field is produced.

point. However, minimum distance to a data cell is a simplistic metric that does not account for the number and spatial configuration of the data cells (*Keller et al., 2015*). In addition, using the minimum distance from data cells $D(P)$ produces a field that has discontinuities along the edges of the discrete Voronoi polygons (Fig. 1B) that are contrary to the property (ii) of the natural neighbour interpolation method that creates surfaces free of any discontinuities. Therefore, the *natural neighbour distance* $\delta$ is presented as a more appropriate measure of distance that incorporates information about the number, spatial distances, and relative positions of the data cells forming the interpolation.

The method to calculate $\delta$ follows a very similar approach to that of calculating the interpolation, and therefore recycles various data structures that are used for the interpolation. For each interpolation cell $c_i$ the Euclidean distances to all data cells are calculated $d_j = d(c_i \rightarrow p_j)$, and then using the Voronoi diagram $V(P)$ these distances are interpolated via nearest neighbour interpolation to produce $D(P)$ that is the distance to the data cells mapped into the discrete Voronoi polygons (Fig. 2A).

The set $V(c_i)$ can be used again to find the set of relevant data cell distances

$$D(c_i) = \{c \in D(P) | c \in V(c_i)\} \tag{7}$$

that will form the basis of the interpolation to that cell (Fig. 2B). The natural neighbour distance is then calculated as

$$\delta(c_i) = \frac{\sum D(c_i)}{\sharp D(c_i)} \tag{8}$$

that is simply the mean value of the distances for the cells in $D(c_i)$. With $\delta$ calculated for all raster cells it becomes evident that unlike minimum distance that contains spatial

discontinuities (Fig. 1B) the natural neighbour distance creates a smooth surface free of any discontinuities (Fig. 2C). Also, the minimum distance is an optimistic measure of distance as it only accounts for the closest data cell, whereas by comparison the distances for $\delta$ are larger as they recognise that the other data cells involved in the interpolation are further away.

### Cross-validation error-distance field

To incorporate $\delta$ into the estimate of error to produce a cross-validation error-distance field, the first step is still a cross-validation process in which each data cell is iteratively withheld and an estimate of the value of the withheld data cell is made with the remaining $n-1$ data cells. However, the absolute error $e = |z_i - \hat{z}_i|$ is now divided by the natural neighbour distance $\delta$ to calculate a rate of error $r$ for each data cell

$$r_i = \frac{|z_i - \hat{z}_i|}{\delta_i} \tag{9}$$

with these rates of error stored so that each data cell becomes $p_i = \{x_i, y_i, z_i, r_i\}$. Then when conducting the natural neighbour interpolation, while estimating the value $\hat{z}$ an estimate of the rate of error $\hat{r}$ can be simultaneously produced (Fig. 3A) and used to produce an error estimate

$$\hat{e}_i = \hat{r}_i \times \delta_i \tag{10}$$

that when estimated for all cells produces a cross-validation error-distance field (Fig. 3B). The cross-validation error-distance field clearly captures information from the rate of error field (Fig. 3A) and the natural neighbour distance field (Fig. 2C) with lower error estimates in areas that have either low rates of error or natural neighbour distances, and higher error estimates in areas that have higher rates of error and/or natural neighbour distances. Therefore, the cross-validation error-distance field captures uncertainty information relating to local variation in both the autocorrelation of the underlying phenomenon field being interpolated and the spatial distribution of the data cells providing data for the interpolation.

## Virtual geography experiments

The discrete natural neighbour interpolation and cross-validation error-distance field algorithms described here were implemented using a Python computational framework (*Pérez, Granger & Hunter, 2011*) using the NumPy (*Van der Walt, Colbert & Varoquaux, 2011*), SciPy (*Virtanen et al., 2020*), and Matplotlib (*Hunter, 2007*) packages. Having proposed a new method, it is sensible to provide an evaluation of how performance varies under different conditions. However, in doing so it is important to remember that interpolation errors result not only from the efficacy of the interpolation method, but also from distribution of data points and the real (but unknown) distribution of the phenomenon field being interpolated (*Willmott & Matsuura, 2006*) that will be unique to each study. Also, what constitutes an acceptable level of interpolation error will also vary between studies. Therefore, the objective here is try and identify simple trends in performance to verify the methods work as would be expected and to provide some basic

**Peer**J Computer Science

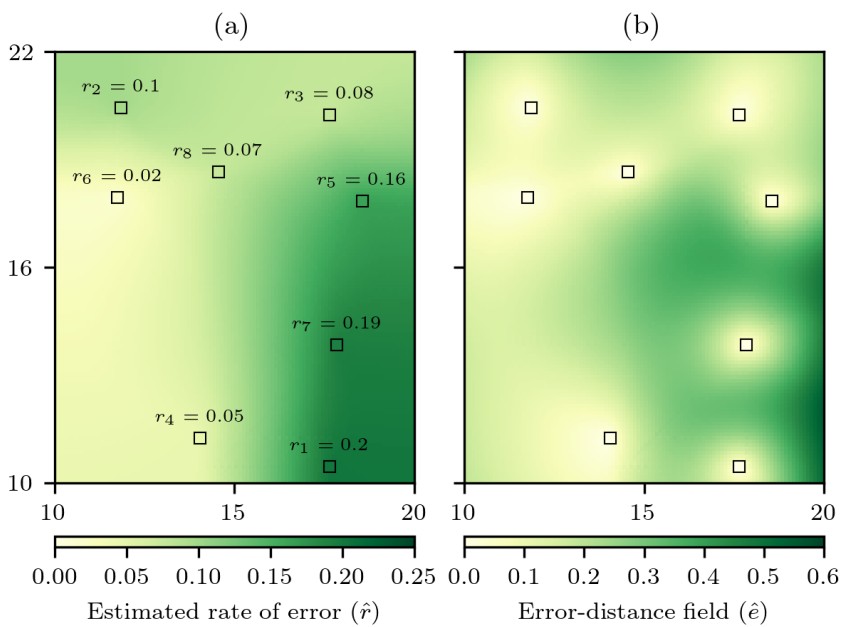

**Figure 3 Computation of the cross-validation error-distance field.** (A) The rate of absolute error for each data cell $r_i$ calculated through cross-validation, and then an estimated rate of absolute error field $\hat{r}$ is produced by natural neighbour interpolation of $r$. (B) The cross-validation error-distance field $\hat{e}$ that is the product of $\hat{r}$ and the natural neighbour distance $\delta$ for each interpolation cell.

information that will help an analyst to make a more detailed assessment of whether interpolation is feasible or not.

To evaluate the effectiveness of the proposed interpolation methods, a series of *in silico* virtual geography experiments were conducted. Virtual geographies are a very useful approach for methodological evaluation as the conditions can be tightly controlled and explored fully. Virtual geographic phenomena fields for grids of $100 \times 120$ cells were created using the NLMpy package (*Etherington, Holland & O'Sullivan, 2015*) implementation of the mid-point displacement fractal algorithm that produces fields representing natural phenomena such as land surfaces (*Fournier, Fussell & Carpenter, 1982*). The spatial-autocorrelation of the values produced by the mid-point displacement method can be controlled by varying the $h$ parameter to produce fields with spatial-autocorrelation that varies from low to high (Fig. 4).

The underlying premise of the experiments was that with random sampling of a virtual geographic phenomenon with actual values $z$ (Fig. 5A), natural neighbour interpolation can be used to produce estimated values $\hat{z}$ (Fig. 5B). The absolute difference between the actual values and the estimated values is the value error $e(\hat{z}) = |\hat{z} - z|$ (Fig. 5C) that will indicate how well the natural neighbour interpolation method works. The value error is also estimated by the cross-validation error-distance field $\hat{e}$ (Fig. 5D), and the absolute difference between the value error $e(\hat{z})$ and the estimated error $\hat{e}$ is the error of errors $e(\hat{e}) = |\hat{e} - e(\hat{z})|$ that indicates how well the proposed cross-validation error-distance field performs (Fig. 5E).

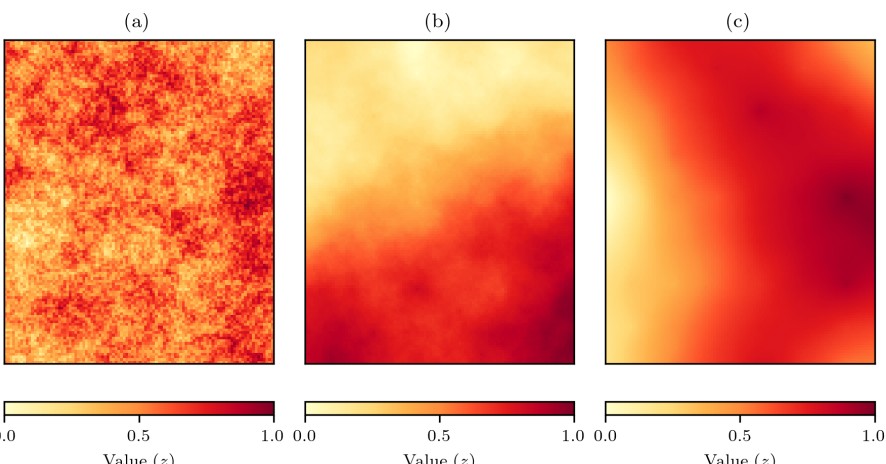

**Figure 4** **Examples of virtual geographic phenomena fields created by the mid-point displacement fractal algorithm.** The spatial-autocorrelation varies from low to high and is controlled by the $h$ parameter that in these examples has been set to (A) $h=0$, (B) $h=1$, and (C) $h=2$.

To summarise the performance of both natural neighbour interpolation and the cross-validation error-distance field, the MAE (Eq. (6)) was calculated for the cells inside and outside of the convex hull of the sampling points for both $e(\hat{z})$ (Fig. 5C) and $e(\hat{e})$ (Fig. 5E). The MAE was chosen as the error statistic as it expresses error in the same units as the variable of interest and is insensitive to the number of cells in the sample (*Willmott & Matsuura, 2006*), which was important here as the convex hull area would vary as a result of the random sampling.

When the spatial-autocorrelation and number of sample points is reduced we would expect a reduction in performance of both the natural neighbour interpolation and the cross-validation error-distance field (Figs. 5A–5E versus Figs. 5F–5J). Therefore, to examine how the natural neighbour methods performed under varying conditions 500 experiments were conducted in which $h$ randomly varied uniformly between 0.0 to 2.0 and $n$ randomly varied uniformly between 10 to 100. The cross-validation MAE was also calculated for each experiment to assess if the cross-validation MAE could be used as an indicator of expected interpolation performance.

## RESULTS

The results from the virtual geography experiments demonstrate that, as would be expected for the cells within the convex hull of the sampling points, the MAE of the value errors $e(\hat{z})$ from the natural neighbour interpolation (Fig. 6A) and error of errors $e(\hat{e})$ from the cross-validation error-distance field (Fig. 6B) reduced as the number of data points $n$ and the spatial-autocorrelation $h$ of the underlying virtual phenomena fields increased. The effect of $h$ was more important, as when $h$ was low or high $n$ did not have much effect on the performance. The importance of $h$ is to be expected as all interpolation methods work on the assumption that the phenomenon being interpolated has sufficient levels of spatial-autocorrelation.

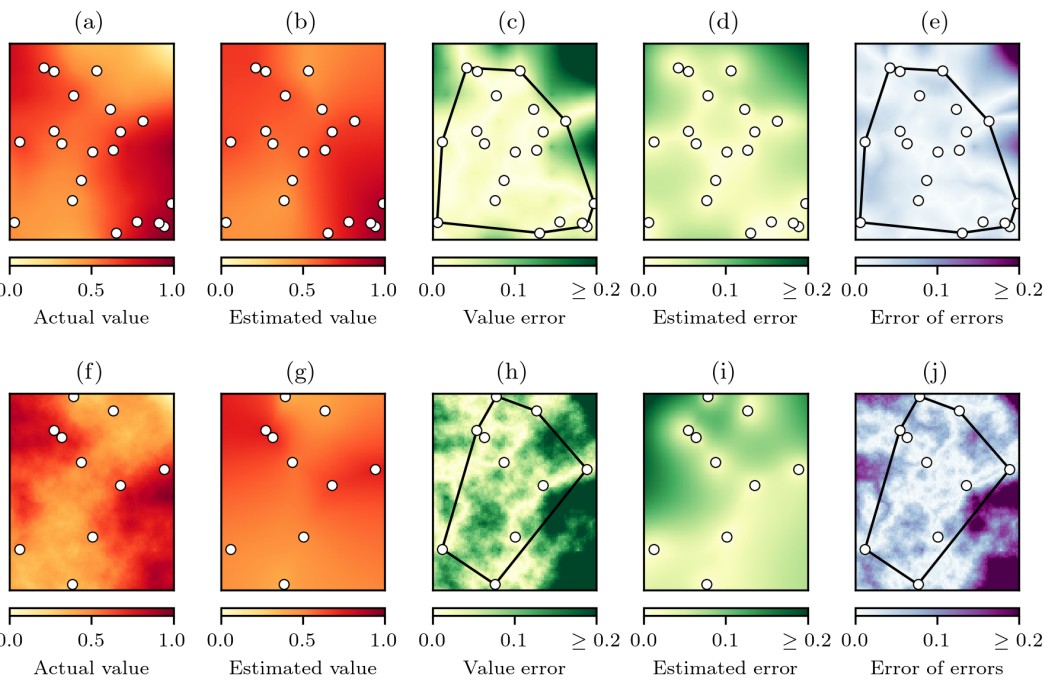

**Figure 5** **The natural neighbour interpolation virtual geography experimental process.** (A) A virtual geography phenomenon field $z$ with spatial-autocorrelation of $h = 2$ and $n = 20$ random sampling points, (B) the resulting natural neighbour interpolation $\hat{z}$ from the sampling points, and (C) value error $e(\hat{z}) = |\hat{z} - z|$. (D) The cross-validation error-distance field estimated error $\hat{e}$ that is also produced during interpolation is then compared to the value error $e(\hat{z})$ to produce (E) the error of errors $e(\hat{e}) = |\hat{e} - e(\hat{z})|$. Interpolation performance as a function of $e(\hat{z})$ and $e(\hat{e})$ was summarised for cells within and outside the convex hull of the sampling points. The same experimental process in (A–E) is replicated in (F–J) for a virtual geography phenomenon field with spatial-autocorrelation of $h = 1$ and $n = 10$ random sampling points, demonstrating a reduction in interpolation performance at lower levels of spatial-autocorrelation and sampling.

There was also a very strong correlation between $e(\hat{z})$ and $e(\hat{e})$ (Fig. 6C) and this similarity of behaviour under different conditions indicates that the cross-validation error-distance field meets the objective of providing a measure of uncertainty that is consistent with all the useful properties of natural neighbour interpolation.

While the results of the virtual geography experiments (Figs. 6A and 6B) indicate that lower average errors can be expected when $n \gtrsim 20$ and $h \gtrsim 1.0$ (Fig. 4B) such criteria cannot be easily applied by an analyst as while $n$ is known $h$ is unknown and in many situations will be hard to guess. Fortunately, while the cross-validation MAE that can always be calculated by an analyst is generally slightly higher than the $e(\hat{z})$ there is still a strong correlation between the two variables (Fig. 6D), and this correlation is extremely useful as it indicates to an analyst the likely levels of $e(\hat{z})$ and therefore $e(\hat{e})$ too.

A comparison of $e(\hat{z})$ and $e(\hat{e})$ inside and outside of the convex hull around the sampling points clearly shows that while the performance follows a similar trend $e(\hat{z})$ and $e(\hat{e})$ can be expected to be higher outside of the convex hull (Figs. 6E–6F).

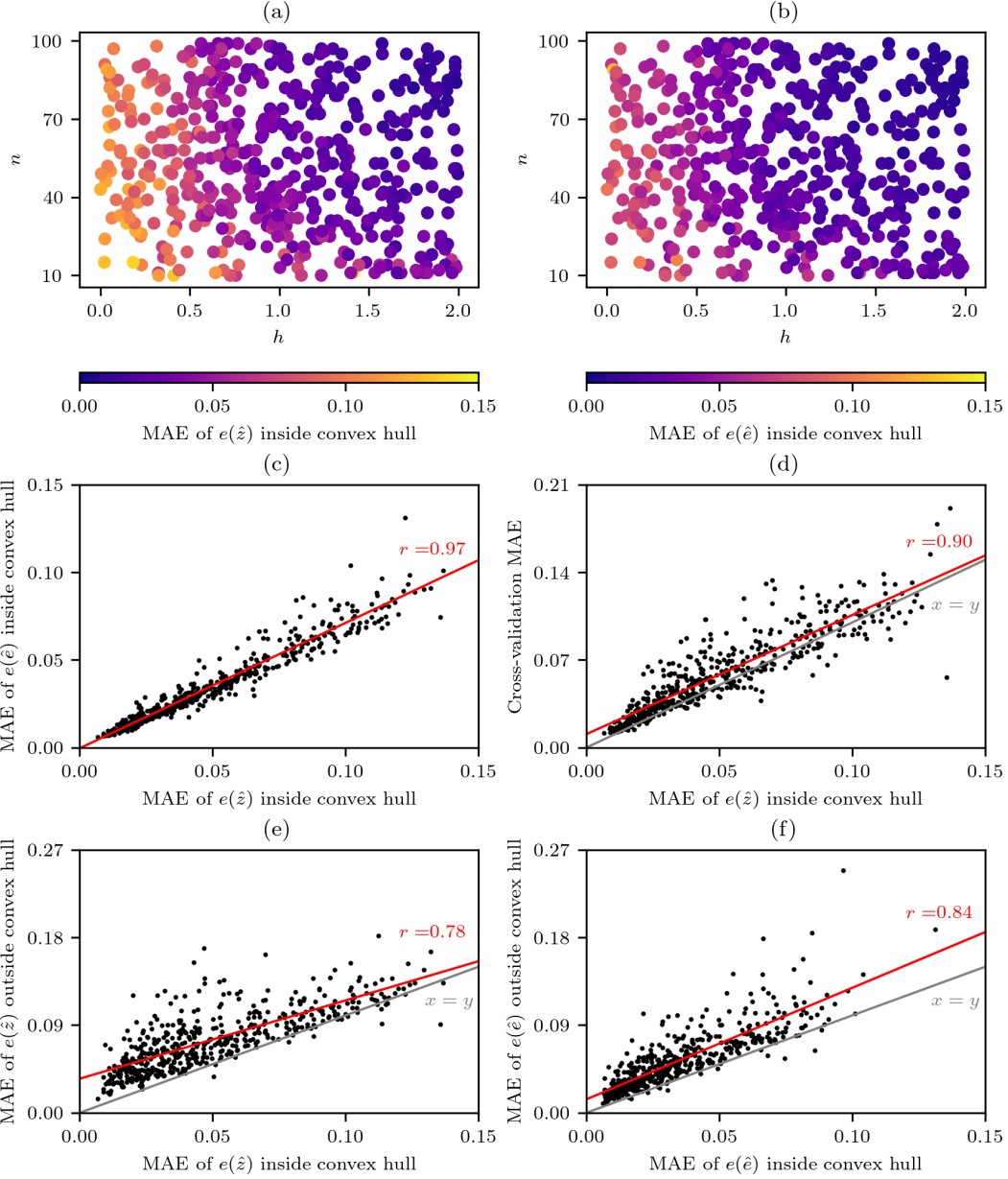

**Figure 6** **Performance of natural neighbour interpolation and cross-validation error-distance fields from 500 virtual geography experiments.** The mean absolute error (MAE) of cells within the convex hull around sampling points for different experimental combinations of the number $n$ of random sampling points and the spatial-autocorrelation $h$ of virtual phenomena fields for (A) the value errors $e(\hat{z})$ from the natural neighbour interpolations and (B) the error of errors $e(\hat{e})$ from the cross-validation error-distance fields that (C) were highly correlated. (D) Comparison of $e(\hat{z})$ and the cross-validation MAE derived from the sampling points. Comparison of interpolation performance inside and outside the convex hull around the sampling points for (E) $e(\hat{z})$ and (F) $e(\hat{e})$.

## DISCUSSION

The virtual geography experiments indicate that under suitable conditions the natural neighbour interpolation field and the cross-validation error-distance field should provide useful estimates of a geographic phenomenon field with associated uncertainty. The fact that the cross-validation error-distance field reflects localised changes in the spatial distribution of both the underlying phenomenon and the point data is particularly useful, and contrasts with other spatial interpolation uncertainty methods such as MAE and RMSE that estimate error using a global approach.

The virtual geography experiments demonstrated that the performance of natural neighbour interpolation will be lower outside of the convex hull around the data points, as is expected (*Watson, 1999*)—although this is also likely to be true of all spatial interpolation techniques as beyond the convex hull interpolation becomes extrapolation. However, we do not suggest that interpolation should be restricted to within the convex hull as there may be occasions where the area of interest may occur slightly outside the convex hull. For example, when interpolating rainfall data from weather stations that are usually sited in settlements, there are likely to be areas of coastline along peninsulas and headlands that will not fall within a convex hull around the weather stations (*Lyra et al., 2018*). Therefore, it is logistically useful that discrete natural neighbour interpolation can produce estimated values beyond the convex hull of the available data points. What is helpful in this context is that the cross-validation error-distance field incorporates information on distance from data points, therefore as interpolations move further beyond the convex hull the error-field should increase to help to guard against erroneous estimates.

However, the responsibility of appropriate use of natural neighbour interpolation still belongs with the spatial analyst who must make decisions about whether interpolation is useful based on their knowledge of: the expected spatial-autocorrelation of the phenomenon being interpolated, the number and distribution of data points, the location of the areas for which interpolations are required, and the magnitude of the estimated errors in relation to the magnitude of the value estimates. And of course, the cross-validation error-distance field only captures uncertainty in the interpolation itself and does not incorporate any uncertainty that may arise from the data itself. While I have argued against the use of the cross-validation MAE as a measure of uncertainty, I would recommend that analysts continue to calculate the cross-validation MAE given its strong correlation with the performance of the natural neighbour interpolation, and therefore the performance of the cross-validation error-distance field too. Analysts can then use the cross-validation MAE as a helpful guide when deciding if interpolation is advisable or not. When doing so it is important to remember that as the cross-validation MAE is based on the use of $n-1$ data cells, the error estimates may be slightly higher than the real errors that would be based on all $n$ data that is ultimately used in the interpolation (*Willmott & Matsuura, 2006*). Therefore, the cross-validation MAE should be seen as a slightly conservative indication of likely interpolation performance.

## CONCLUSION

For those researchers for whom natural neighbour interpolation is the best interpolation option, the cross-validation error-distance field method presented provides a way to assess the uncertainty associated with natural neighbour interpolations that is consistent with the useful properties of natural neighbour interpolation. While the cross-validation error-distance method has been described here in the context of discrete natural neighbour interpolation, there is no reason why this same approach could not be applied to geometric natural neighbour interpolation as well. Discrete natural neighbour interpolation has been implemented here in two-dimensional space for ease of visualisation, but the method will generalise to higher dimensions (*Park et al., 2006*) and in principle I cannot see any reason why the uncertainty method presented could not also be applied in higher dimensions by those who wish to do so. The approach could easily be adapted to other interpolation methods, as all that is required is a measure of weighted distances to the data points creating the interpolation. Given the promise of the algorithm, and to encourage its use and development, the Python code used to generate the examples presented is freely available under the permissive MIT License as supplementary material.

### Funding

This work was supported by the Strategic Science Investment Funding for Crown Research Institutes from the New Zealand Ministry of Business, Innovation and Employment's Science and Innovation Group. The funders had no role in study design, data collection and analysis, decision to publish, or preparation of the manuscript.

### Grant Disclosures

The following grant information was disclosed by the author:
Strategic Science Investment Funding for Crown Research Institutes from the New Zealand Ministry of Business, Innovation and Employment's Science and Innovation Group.

### Competing Interests

Thomas R. Etherington is employed by Manaaki Whenua-Landcare Research, and declares that there are no competing interests.

### Author Contributions

- Thomas R. Etherington conceived and designed the experiments, performed the experiments, analyzed the data, prepared figures and/or tables, authored or reviewed drafts of the paper, and approved the final draft.

### Data Availability

Python scripts to reproduce the examples, virtual experiments, and figures are available as a Supplementary File.

## Supplemental Information

Supplemental information for this article can be found online at http://dx.doi.org/10.7717/peerj-cs.282#supplemental-information.

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
