# Peer review of "Discrete natural neighbour interpolation with uncertainty using cross-validation error-distance fields"

_PeerJ Computer Science, doi:10.7717/peerj-cs.282_

## Round 0.1 · original submission · Minor Revisions

· Academic Editor

Minor Revisions

The three reviewers provided valuable suggestions while recommending the paper for publication with some minor revisions. Please carefully address all the comments, especially the following:

• Reference the other spatial interpolation uncertainty methods which estimate error using a global model fitted to all the data simultaneously (lines 221 to 222);
• Have the correlation coefficient between modeled and actual prediction errors computed and plotted versus n and h, as for the MAE;
• If possible, explore how well the approach performs in extrapolation situation;
• Can the authors construct a dimensionless quantity (perhaps number of samplers divided by area multiplied by autocorrelation distance squared) and look at performance as a function of that?
• Consider including a measure of bias for the estimated error.

Reviewer 1 ·

Basic reporting

• Clear and unambiguous, professional English used throughout.
The article is written in English and uses clear, unambiguous, technically correct text. However, sometimes the manuscript is written in first person. I suggest avoiding the use of “I” or “We”. Instead writing “I present an approach” use “this article presents an approach” (example of the lines 62/63).

• Literature references, sufficient field background/context provided.
The article includes sufficient introduction and background to demonstrate how the work fits into the broader field of knowledge.

• Professional article structure, figures, tables. Raw data shared.
The structure of the article is conform to an acceptable format of ‘standard sections’.
Figures are relevant to the content of the article, of sufficient resolution, and appropriately described and labeled.
All appropriate raw data are available in accordance with our Data Sharing policy.

• Self-contained with relevant results to hypotheses.
The submission is ‘self-contained,’ represents an appropriate ‘unit of publication’, and includes all results relevant to the hypothesis.

Experimental design

• Original primary research within Aims and Scope of the journal.
The article is strongly related do geoprocessing and GIS, which are not presented at the Journal`s Aims & Scopes. However, GIS is an useful tools to Environmental Sciences. I suggest the author to write an paragraph that demonstrates the strong relationship of interpolation techniques with the environmental sciences.

• Research question well defined, relevant & meaningful. It is stated how research fills an identified knowledge gap.
The submission clearly defines the research question, which is relevant and meaningful. The knowledge gap being investigated is identified, and statements are made as to how the study contributes to filling that gap.

• Rigorous investigation performed to a high technical & ethical standard.
The investigation was conducted rigorously and to a high technical standard.

• Methods described with sufficient detail & information to replicate.
Methods were described with sufficient information to be reproducible by another investigator.

Validity of the findings

• Impact and novelty not assessed. Negative/inconclusive results accepted. Meaningful replication encouraged where rationale & benefit to literature is clearly stated.
The article is not a ‘pointless’ repetition of well known, widely accepted results.

• All underlying data have been provided; they are robust, statistically sound, & controlled.
The data on which the conclusions are were made available in an acceptable discipline-specific repository. The data is robust, statistically sound, and controlled.

• Conclusions are well stated, linked to original research question & limited to supporting results.
The conclusions are appropriately stated, also connected to the original question investigated, and are limited to those supported by the results.

Additional comments

Please, consider to add the equations for MAE (lines 189 and 191) and MAD (line 200).
Please reference the other spatial interpolation uncertainty methods which estimate error using a global model fitted to all the data simultaneously (lines 221 to 222).

Reviewer 2 ·

Basic reporting

This paper presents the use of cross-validation error-distance fields to assess the uncertainty attached to estimates provided by natural neighbour interpolation. The paper appears technically sound and is well illustrated; the manuscript however includes multiple typos (e.g., lines 146, 149, 208) and would need a careful reading.

Experimental design

My main comment relates to the virtual geography experiments:
1) the use of MAE as performance criterion simply indicates that the average prediction errors decrease with the number of data and the strength of the spatial autocorrelation; it doesn’t directly tell me whether the error distance field accurately portrays where predictions errors are large or small. I would recommend that the correlation coefficient between modeled and actual prediction errors be computed and plotted versus n and h, as for the MAE.
2) the virtual geography experiments were limited to the convex hull around the data points and it would be useful to explore how well the approach performs in extrapolation situation.

Validity of the findings

The virtual geography experiments were limited to the convex hull around the data points and it would be useful to explore how well the approach performs in extrapolation situation.

Additional comments

This paper presents the use of cross-validation error-distance fields to assess the uncertainty attached to estimates provided by natural neighbour interpolation. The paper appears technically sound and is well illustrated; the manuscript however includes multiple typos (e.g., lines 146, 149, 208) and would need a careful reading. My main comment relates to the virtual geography experiments:
1) the use of MAE as performance criterion simply indicates that the average prediction errors decrease with the number of data and the strength of the spatial autocorrelation; it doesn’t directly tell me whether the error distance field accurately portrays where predictions errors are large or small. I would recommend that the correlation coefficient between modeled and actual prediction errors be computed and plotted versus n and h, as for the MAE.
2) the virtual geography experiments were limited to the convex hull around the data points and it would be useful to explore how well the approach performs in extrapolation situation.

·

Basic reporting

The paper is well written and structured clearly.
Below are listed a few minor points in roughly the order in which they occur in the text.

Line 146 – “recognise that the over data cells” should be “recognize that the other data cells”
Line 149 Sentence should start with “To” rather than “The”
Line 211 spelling of neighbor.

Figure 6. Since the MAE seems to be more interesting than the MAD, it might be preferable to plot the MAE as the colors and the MAD on the z axis. It is easier to judge the value of the colors than the value along the z axis.

Figure 6. Are the numbers on the color scale correct? In experiments.py, colors are broken into values from 0 to 0.02 (rather than 0 to 0.2) and listing of values in nniMAEMAD variable show values up to about 0.02 rather than 0.2. Tick values on legend are set separately rather than being tied to values in the array.

Supplemental material.
I was able to download and run the python scripts provided with little trouble.
I had to comment out the lines pertaining to the use of LaTex (see below). These lines were not critical to producing the figures and the python scripts reproduced the figures in the manuscript without them.

# Check if LaTeX is installed to plot with consistent math font
from distutils.spawn import find_executable
if find_executable('latex'):
mpl.pyplot.rc('text', usetex=True)

Including these lines generated the following error.
!Latex Error: File `typelec.sty’ not found

In experiment.py line 116 changed
Dnni.NNI(xyz,C) to dnni.nni(xyc,c)
As the discreteNaturalNeighbourInterpolation.py has a function called nni, not NNI.

Experimental design

The evaluation is a bit cursory and a few suggestions are given for expanding it.

The authors run 100 experiments for each combination of h (amount of correlation) and n (number of samplers). Each experiment has a different underlying simulated geometry and placement of samplers as both are generated with random processes. The MAE is computed for each of the 100 experiments and a median MAE and a median absolute deviation, MAD, are reported in Figure 6. It isn’t clear to me why the results from the 100 experiments are not just combined to calculate one value of MAE. There is no discussion provided about the MAD. MAD seems to be most strongly dependent on n (rather than h) which could be because the convex hull will tend to include less points for small n and thus each experiment represents a smaller sample size and so there will be more variation between the samples. Some discussion is needed if this metric is included.

Line 210-211 ‘indicate that the cross-validation error-distance field provides more accurate and precise estimates than the natural neighbour interpolation field” The range of values was 0 to 1. Whereas the range of errors was about 0 to 0.25. Consequently this statement is misleading as MAE is being used to compare datasets with different scales.

Lines 212-215. The summary is not particularly useful as is. All units are arbitrary which makes the evaluation dataset general but statements such as “when the number of data points n>= 40…. And spatial autocorrelation h>=1” are not useful in any other context. Can the authors construct a dimensionless quantity (perhaps number of samplers divided by area multiplied by autocorrelation distance squared) and look at performance as a function of that?

Line 214-215 “should perform well”. The criteria for good performance should be stated.

Lines 223-225 “… the error estimates may be slightly higher than the real errors…” It may be worthwhile to include a measure of bias for the estimated error. A quick look at the full distribution of one of the experiments error difference (errorDifference variable in experiments.py) does show a distribution that is skewed positive. On the other hand the distribution of the actual error seems symmetric about 0, showing that the interpolation method itself does not produce estimated values which are biased.

Validity of the findings

no comment.

Additional comments

The manuscript describes a method for estimating uncertainty in values obtained from natural neighbor interpolation. The focus is on use for two-dimensional geospatial data as the evaluation dataset is produced using an algorithm designed to produce a simulate landscapes. However, the technique itself seems quite general. The background and methodology are clearly described. The authors provide all code necessary to recreate their results in the supplemental material. The code is well documented and easy to run for someone with some knowledge of python. The evaluation using the simulated landscapes is adequate to show that the method is of interest. The evaluation is a bit cursory and a few suggestions are given for expanding it. I recommend the paper for publication with some minor revisions.

---

## Round 0.2 · accepted · Accept

· Academic Editor

Accept

PeerJ staff will arrange to transfer the submission to PeerJ Computer Science.